# Immunological Cross-Protection between Different Rabbit Hemorrhagic Disease Viruses—Implications for Rabbit Biocontrol and Vaccine Development

**DOI:** 10.3390/vaccines10050666

**Published:** 2022-04-22

**Authors:** Tiffany W. O’Connor, Andrew J. Read, Robyn N. Hall, Tanja Strive, Peter D. Kirkland

**Affiliations:** 1Virology Laboratory, Elizabeth Macarthur Agricultural Institute, NSW Department of Primary Industries, Menangle, NSW 2568, Australiaandrew.j.read@dpi.nsw.gov.au (A.J.R.); 2Health & Biosecurity, Commonwealth Scientific and Industrial Research Organisation, Acton, ACT 2601, Australia; robyn.hall@csiro.au (R.N.H.); tanja.strive@csiro.au (T.S.); 3Centre for Invasive Species Solutions, Bruce, ACT 2617, Australia

**Keywords:** lagovirus, rabbit hemorrhagic disease virus, RHDV1, RHDV2, rabbit calicivirus, cross-protection, immunity, biocontrol, vaccine

## Abstract

The use of rabbit hemorrhagic disease virus (RHDV) as a biocontrol agent to control feral rabbit populations in Australia, in combination with circulating endemic strains, provides a unique environment to observe the interactions between different lagoviruses competing for the same host. Following the arrival of RHDV2 (GI.2) in Australia, it became necessary to investigate the potential for immunological cross-protection between different variants, and the implications of this for biocontrol programs and vaccine development. Laboratory rabbits of various immune status—(1) rabbits with no detectable immunity against RHDV; (2) rabbits with experimentally acquired immunity after laboratory challenge; (3) rabbits immunised with a GI.2-specific or a multivalent RHDV inactivated virus prototype vaccine; or (4) rabbits with naturally acquired immunity—were challenged with one of three different RHDV variants (GI.1c, GI.1a or GI.2). The degree of cross-protection observed in immune rabbits was associated with the variant used for challenge, infectious dose of the virus and age, or time since acquisition of the immunity, at challenge. The immune status of feral rabbit populations should be determined prior to intentional RHDV release because of the high survival proportions in rabbits with pre-existing immunity. In addition, to protect domestic rabbits in Australia, a multivalent RHDV vaccine should be considered because of the limited cross-protection observed in rabbits given monovalent vaccines.

## 1. Introduction

Rabbit hemorrhagic disease virus (RHDV), a member of the genus Lagovirus in the family Caliciviridae, is used effectively as a biocontrol agent in Australia to reduce the significant environmental and economic impact of feral European rabbit (*Oryctolagus cuniculus*) populations [1]. Specifically, viruses within the GI.1 genotype are deliberately released as part of these control efforts, with a GI.1c (classical RHDV) virus in use since 1995 and a GI.1a (RHDV1a) virus in use since 2017 [2,3,4]. In addition to these intentionally released lagoviruses, a non-pathogenic rabbit calicivirus, GI.4 (RCV-A1), is endemic, while another virus, GI.2 (RHDV2), is currently the dominant pathogenic variant across Australia [5,6]. Since the first detection of GI.2 in Australia, the original GI.2 variant (GI.1bP-GI.2) has been replaced by several other naturally occurring GI.2 recombinants, all of which comprise nearly identical VP60 capsid proteins [5,7,8]. The deliberate release of RHDV, in combination with these naturally circulating lagoviruses in Australia, provides a unique environment to observe the evolving interactions between different lagoviruses. These viruses compete for the same host, and this has implications for both the biological control of feral rabbits as well as appropriate vaccine protection for domestic rabbits in Australia.

Rabbit biocontrol programs are very successful in Australia, with the resulting feral population decline delivering an estimated $1 billion AUD in economic and environmental benefits annually [1,3,9]. Nevertheless, both intentionally released and naturally circulating lagoviruses also have significant impacts on domestic rabbits. In a susceptible rabbit, infection with RHDV causes a fatal, fulminant necrotising hepatitis and disseminated intravascular coagulation [10]. Currently, there is one lagovirus vaccine registered in Australia for domestic rabbits (Cylap^®^ RCD, Zoetis, NJ, USA). This vaccine offers complete protection against both the GI.1c and GI.1a variants used in biocontrol programs [11], but GI.1 vaccines are largely ineffective against GI.2 viruses [12]. In addition, this inactivated vaccine is imported and there can be supply shortages, leaving domestic rabbits in Australia vulnerable. Several GI.2 replication-competent, viral-vectored vaccines are manufactured overseas, but these are not suitable for the Australian context. The theoretical transmission of these vaccine viruses into the feral rabbit population could significantly reduce the economic savings and environmental benefits of rabbit control programs [13]. To date, European inactivated vaccines have not been imported and registered in Australia, possibly due to a small market [14]. Consequently, there is a need for a suitable Australian vaccine to protect domestic rabbits against unintentional infection from all circulating pathogenic lagoviruses (GI.1 and GI.2).

Cross-protection between different RHDV variants was investigated previously in the context of both natural immunity acquired after challenge and vaccinal immunity [11,15]. Australian rabbits previously exposed to GI.4 were partially and transiently protected against GI.1c challenge [16], and lead to the investigation of a GI.1a variant (Korean origin) as an additional biocontrol agent to overcome this cross-protection [4]. There is limited immunological cross-protection between genotypes GI.1a and GI.2 [15,17]. This lack of cross-protection is promising for the selection of a GI.2 variant, GI.1bP-GI.2, as the next biocontrol agent, where this virus may overcome existing immunity in a feral rabbit population. However, limited cross-protection is problematic for vaccine development as an effective vaccine must protect a domestic Australian rabbit against GI.1a, GI.1c and GI.2.

The objectives of this research were to establish whether prior immunity to different RHDV genotypes could offer cross-protection against GI.2. Three types of immunity were investigated: firstly, rabbits with experimentally acquired immunity after primary infection with the current GI.1a biocontrol variant or the non-pathogenic GI.4 virus (Figure 1A); secondly, rabbits vaccinated with either a prototype GI.2-specific vaccine or a multivalent RHDV vaccine (Figure 1B); and, finally, rabbits that serendipitously had naturally acquired immunity to GI.1 or GI.2 (Figure 1C). Rabbits from these three groups were challenged with RHDV variants from either the GI.1a, GI.1c or GI.2 genotypes. The interactions between the different Australian lagoviruses explored here have implications for selection and delivery of virus strains for biocontrol purposes and vaccine composition for optimal protection.

## 2. Materials and Methods

### 2.1. Animals and Experimental Design

Young domestic rabbits with various immunity and infection statuses were obtained from three geographically distinct regions in Australia. Both the cross-protection experiment on experimentally acquired immunity (Figure 1A, Table 1) and vaccinal immunity (Figure 1B) were planned. For these experiments, sample sizes were estimated based on demonstrating a difference in survival between groups. The investigation of cross-protection with rabbits with naturally acquired immunity (Figure 1C, Table 1) was serendipitous and sample sizes were determined by the number of available rabbits. All rabbits were randomly allocated into their respective treatment groups. All procedures involving animals were carried out in accordance with the “Australian Code of Practice for the Care and Use of Animals for Scientific Purposes” and approved by the NSW Department of Primary Industries, Elizabeth Macarthur Agricultural Institute (EMAI, Menangle, Australial), Animal Ethics Committee (M20-02). Challenge studies were performed at EMAI.

### 2.2. Cross-Protection in Rabbits with Experimentally Acquired Immunity

New Zealand White rabbits of both sexes were obtained from a colony known to be free of benign and pathogenic lagoviruses. Prior to both experimental challenges, all rabbits were confirmed to be seronegative to Australian lagoviruses by ELISA [18].

To obtain rabbits with GI.1a immunity, five-week-old rabbits were orally inoculated with 100 ‘50% Rabbit Infectious Doses’ (RID_50_) of a commercially available GI.1a infectious virus preparation (GenBank acc. Number MF598301.1) [11]. The RID_50_ of the GI.1a challenge was titrated as part of the quality control tests for commercial preparations of RHDV. To obtain rabbits with GI.4c immunity, five-week-old rabbits were given a 1 mL oral dose (with an estimated 2300 genome copies) of a clarified homogenate of duodenal tissue from a GI.4c-infected rabbit (GenBank acc. Number KX357705.1). Titration to determine RID50 of GI.4c was not performed. Therefore, ‘genome copies’ is used as an in vitro approximate to quantify the GI.4c challenge.

After initial challenge, all rabbits seroconverted. To simulate infection against peak immunity, rabbits were then challenged with GI.2 within five weeks of initial infection [16]. Thus, 39 days after infection with GI.1a or GI.4c viruses (i.e., at 12 weeks of age), rabbits were challenged *per os* (PO) with 50 RID_50_ of GI.1bP-GI.2 (GenBank acc. Number MW467791) (Figure 1A).

### 2.3. Cross-Protection in Rabbits Vaccinated with a Prototype Vaccine

Rabbits from a commercial breeding facility that were seronegative to GI.1 and GI.2 were vaccinated with a single dose of either a prototype, inactivated GI.2-specific vaccine or a prototype inactivated multivalent RHDV vaccine (with GI.1a, GI.1c and GI.2 components). Rabbits were vaccinated by subcutaneous injection at 10–12 weeks of age and then challenged PO with a heterologous lagovirus 28 days after vaccination (Figure 1B).

### 2.4. Cross-Protection in Rabbits with Naturally Acquired Immunity

Rabbits with naturally acquired immunity to lagoviruses were obtained from three commercial rabbit farms in Australia. These rabbits were initially obtained for other purposes. Following the detection of anti-GI.1 or anti-GI.2 antibodies based on serology, or GI.2 disease, these rabbits were then serendipitously selected for inclusion in this study. Although rabbits were not tested for anti-GI.4 antibodies, given the widespread distribution of this virus in Australian rabbit farms, these rabbits could possibly also have had anti-GI.4 antibodies [16]. New Zealand White rabbits from one farm (farm 1) were unvaccinated but were seropositive to GI.1. This was likely to be due to either previous non-lethal exposure to a GI.1c or GI.1a virus or persisting maternal antibodies. Most of these rabbits with GI.1 immunity were challenged PO at 12 weeks of age with GI.2 virus, while some were retained and challenged PO at 33 weeks of age to assess if this immunity waned over time (Figure 1C). New Zealand White rabbits from another farm (farm 2) were unvaccinated but were seropositive to GI.2. The origin of this antibody was likely to be either previous non-lethal exposure to a GI.2 virus or persisting maternal antibodies. These rabbits with GI.2 immunity were challenged PO at 12 weeks of age (Figure 1C). From a third farm (farm 3), New Zealand White cross Flemish Giant rabbits were unvaccinated and within 48 h of arrival at EMAI, sudden deaths occurred in this group. Molecular testing revealed infection with a GI.4cP-GI.2 (4c-recombinant) virus, a variant that was (at the time of this study) endemic to the location of this third farm and not previously detected in the region where EMAI is located (GenBank acc. Number MW460156) [19]. One month after the last signs of disease, surviving rabbits from this GI.4cP-GI.2 infection were challenged PO with a GI.1a or a GI.2 variant (GI.1bP-GI.2) (Figure 1C).

### 2.5. Virus Challenge

To mimic a natural route of infection, all rabbits were challenged by the oral administration of a 1 mL volume of virus solution. To assess cross-protection between circulating Australian lagoviruses, rabbits were challenged with various doses (Table 1) of (1) the GI.1c biocontrol preparation (GenBank acc. Number KT344772.1) (EMAI, Menangle, Australia); (2) the GI.1a biocontrol preparation (EMAI, Menangle, Australia); or (3) a GI.2 (GI.1bP-GI.2) preparation [20].

Initially, a dose of 50 RID_50_ was selected as the challenge dose for rabbits with experimentally or naturally acquired immunity. This was based on an estimate of the amount of RHDV a rabbit may be exposed to during a control program [21]. However, to assess whether responses were dose dependent, in other experiments, this was increased to 150 RID_50_ (low dose in Figure 1C) or 1500 RID_50_ (high dose in Figure 1C). Additionally, an extreme dose of 150,000 RID_50_ was used to determine if this could overcome natural immunity to a different GI.2 variant (that is, a GI.1bP-GI.2 challenge of rabbits with natural GI.4cP-GI.2 immunity). All vaccinated rabbits were given a high 1500 RID_50_ PO dose (Figure 1B) as a routine and substantial demonstration of protection.

### 2.6. Vaccine

Prototype, inactivated, whole-virus vaccines were produced by inoculating naive rabbits with a virus of either the GI.1a, GI.1c or GI.2 genotype. In short, a semi-purified preparation of each virus was then prepared from liver homogenates and inactivated using binary ethylenimine [22]. The antigen preparation, measured in hemagglutination (HA) units per mL [23], was diluted to the desired concentration in phosphate buffered saline and adjuvanted with an equal volume of Montanide ISA-201 (Seppic, Courbevoie, France). Rabbits were injected subcutaneously with either 1 mL of the G1.2-specific vaccine (representing a 100 HA dose), or 0.5 mL of the multivalent RHDV vaccine (consisting of a 32 HA dose of each genotype—GI.1a, GI.1c and GI.2).

### 2.7. Monitoring, Sample Collection

All rabbits were group housed pre-challenge and then held individually in separated cages in climate-controlled, biosafety level 2 rooms during their challenge period to avoid cross-infection between animals. Additionally, experiments were staggered to allow adequate decontamination and fumigation between challenges to avoid accidental cross-infection.

To establish the immune status to RHDV, serum samples were collected at several time points from all rabbits: (1) on arrival; (2) prior to vaccination or intentional exposure to RHDV to obtain natural immunity; (3) prior to challenge; and (4) after euthanasia.

Rabbits were monitored twice daily during the seven-day or fourteen-day challenge period for the development of clinical signs of RHDV (such as sudden death, inappetence, reduced water intake, and reduced faecal output). While it was anticipated that moribund rabbits would be euthanised with an overdose of intravenous barbiturate, euthanasia was not required during the challenge period (i.e., all rabbits that succumbed to RHDV infection in this study died peracutely). All surviving rabbits were humanely killed after this challenge period with an overdose of intravenous barbiturate; however, the rabbits with immunity to GI.2 that survived heterologous challenge were repurposed into other research projects.

### 2.8. Data Analyses

Survival analyses was conducted with R Version 4.0.4 [24]. Packages used for these experiments include survminer [25] and survival [26]. A statistical difference between groups was considered to be significant if the log-rank test gave a *p*-value less than 0.05.

### 2.9. Serological Testing

Three specific ELISAs were used to detect antibodies against GI.1, GI.4 and GI.2 to determine the immune status of rabbits prior to challenge. Details of GI.1 and GI.4 antibody ELISAs are as published [27,28]. A blocking ELISA for GI.2 antibodies, similar to that reported in a previous publication [18], was developed with polyclonal RHDV rabbit anti-sera as the capture antibody, semi-purified GI.2 antigen, and a conjugated GI.2-specific monoclonal antibody (4H12, supplied from IZSLER, Brescia, Italy). The assay was validated on serum samples collected from known seronegative and seropositive rabbits. The cut-off point for a positive antibody result was set at a percentage inhibition of greater than 60%.

To help clarify the immune status of individual rabbits, where cross-reactivity between the different antigens was suspected, samples were also tested in three separate hemagglutination inhibition (HI) tests against GI.1a, GI.1c and GI.2 antigens [23].

### 2.10. Molecular Testing

The presence of viral RNA in swabs of the cut surface of livers was determined using reverse-transcription quantitative PCR (RT-qPCR) assays specific to GI.1, GI.2 and GI.4c [6,29,30]. Details of the nucleic acid extraction and PCR amplification methods and equipment used follow those described previously [31].

## 3. Results

### 3.1. Survival of Seronegative Rabbits Following Various RHDV Challenges

Following RHDV challenge, all seronegative rabbits challenged with GI.1a or GI.1c died, and 7/8 challenged with GI.2 died within 14 days of infection (Table 1, Figure 2). The surviving GI.2-infected rabbit (challenged with a low (50 RID_50_) dose of GI.2) remained seronegative after challenge and GI.2 was not detected by RT-qPCR testing on collected liver swab, indicating that it did not become infected. No seronegative rabbits challenged with a 1500 RID_50_ dose of GI.2 (0/12), GI.1a (0/12) or GI.1c (0/12) survived, with all succumbing to infection within 4 days post challenge (dpc), 6 dpc and 8 dpc, respectively. Median survival for a 50 RID_50_ dose of GI.2 was 4 dpc, compared to 3 dpc for a 1500 RID_50_ dose of GI.2. Median survival times for a 1500 RID_50_ dose of GI.1a and GI.1c were 3.5 and 3 dpc, respectively. These differences were not statistically significant (*p* = 0.3).

### 3.2. Survival of Rabbits with Experimentally Acquired Immunity Following GI.2 Challenge

Following a GI.2 challenge of 50 RID_50_, all (7/7) rabbits with GI.1a acquired immunity survived, as did 7/9 rabbits with GI.4c acquired immunity (Table 1, Figure 3). The survival probability was not statistically different between the two variants (*p* = 0.2). Notably, there was strong evidence for cross-protection against GI.2 challenge for both variants, with a statistically significant survival probability compared to seronegative controls (*p* < 0.0001) given the same 50 RID_50_ GI.2 challenge.

### 3.3. Survival of Vaccinated Rabbits Following Various RHDV Challenges

The GI.2-specific vaccine afforded complete protection against GI.2 challenge (Table 2, Figure 4C), with all nine rabbits (9/9) surviving a homologous challenge of 1500 RID_50_. No GI.2-vaccinated rabbits survived infection with either 1500 RID_50_ of GI.1a (0/6) (Figure 4A) or 1500 RID_50_ of GI.1c (0/6) (Figure 4B). All rabbits that had received the multivalent vaccine survived infection against GI.1a (6/6), GI.1 (6/6) and GI.2 (6/6) challenge of 1500 RID_50_ (Table 2, Figure 4D–F).

### 3.4. Survival of Rabbits with Naturally Acquired Immunity Following Various RHDV Challenges

For rabbits from farm 1, with naturally acquired GI.1 immunity, cross-protection was significantly associated with the variant used for challenge (*p* = 0.002). That is, at 33 weeks of age and when given 1500 RID_50_, all GI.1-immune rabbits survived challenge with GI.1a (7/7) but only one survived challenge with GI.2 (1/7) (Table 1, Figure 5A).

There was a trend towards a dose-dependent difference in survival following GI.2 challenge in rabbits with GI.1 immunity. At 12 weeks of age, when given GI.2, nine rabbits (9/10) given 150 RID_50_ survived challenge but only five rabbits (5/9) given 1500 RID_50_ survived challenge (Table 1, Figure 5A). However, this was not statistically significant (*p* = 0.08).

The cross-protection against GI.2 challenge in rabbits with GI.1 immunity was found to reduce with age and/or time since acquisition of GI.1 immunity. When given the same 1500 RID_50_ dose of GI.2, at 12 weeks of age, five rabbits (5/9) survived challenge; but at 33 weeks of age, only one rabbit (1/7) survived (Table 1, Figure 5A). This difference between age groups was statistically difference (*p* = 0.03).

Rabbits from farm 2, with naturally acquired GI.2 immunity, showed varying degrees of cross-protection against infection with 1500 RID_50_ of either GI.1a, GI.1c or GI.2. One rabbit survived GI.1a challenge (1/4), two rabbits survived GI.1c challenge (2/3) and three rabbits survived GI.2 challenge (3/4) (Table 1, Figure 5B). There was no statistical difference in the observed survival between these different challenges (*p* = 0.3); however, sample sizes were small in this experiment due to the serendipitous nature of acquiring these rabbits. Future robust studies to assess the relationship between GI.2 immunity and heterologous RHDV challenges should be undertaken.

Rabbits from farm 3, that had survived an RHDV (GI.4cP-GI.2) outbreak, had complete cross-protection against a heterologous GI.2 (GI.1bP-GI.2) challenge, with all (13/13) rabbits surviving a 150,000 RID_50_ dose (Table 1, Figure 5C). Moderate cross-protection was observed against GI.1a, with nine rabbits (9/12) surviving a 150 RID_50_ dose challenge and nine rabbits (9/10) surviving a 1500 RID_50_ challenge. There was no statistical difference in the observed survival between these different challenge doses (*p* = 0.7).

## 4. Discussion

Immunological cross-protection between GI.1a, GI.1c and GI.2 lagoviruses has implications for both the use of these viruses as biocontrol agents in feral rabbit populations and the composition of vaccines to provide optimal protection for domestic rabbits in Australia. Here, we present a series of experimental studies in rabbits with various immune statuses and their subsequent response to challenge to Australian lagoviruses.

As expected, all seronegative rabbits succumbed to disease following infection with GI.1c, GI.1a, or GI.2 viruses, demonstrating a very high case fatality rate for all viruses investigated. This is consistent with previous virulence studies with these variants [11,20]. In contrast, we observed significant immunological cross-protection in rabbits with either experimentally or naturally acquired immunity. The degree of cross-protection observed in these immune rabbits was associated with the variant given, the virus dose and the time between the challenge and the previous infection. This is reminiscent of previous reports on the partial cross-protection to GI.1 challenge conveyed by GI.4 immunity, which rapidly declined within weeks following the GI.4 infection [16]. Overall, our findings contrast with previous reports in Europe where there appears to be more limited cross-protection between variants [15,17], as well as reports of GI.1-immune rabbits succumbing to GI.2 in Australia [32], although the time between infections with the two viruses was unknown in that study.

The origin of the naturally acquired immunity for the rabbits from farm 1, with detectable GI.1 antibodies, or for the rabbits from farm 2, with detectable GI.2 antibodies, remains unknown. The GI.1 ELISA is unable to distinguish between GI.1a or GI.1c antibodies, and between antibodies derived from natural infection or GI.1 vaccination. The known presence of GI.4 virus circulating at farm 2 also complicates the serological picture in these rabbits. Rabbits from farm 3 arrived at EMAI unvaccinated and infected with GI.4cP-GI.2. Therefore, the origins of their immunity may be assumed to be a consequence of surviving GI.4cP-GI.2 infection.

The source of the naturally acquired immunity may be due to maternal antibodies. A breeding doe may confer maternal antibodies to her kittens for up to one year after RHDV vaccination [33]. The rabbits from farm 1 were acquired from does vaccinated against GI.1 within one month of their birth. For GI.1, the duration of the protective effects from maternal immunity after infection varies according to the antibody titre of the doe, but was not found to persist beyond approximately 14 weeks of age [34]. The rabbits from farm 2 were acquired from does that survived GI.2 within the previous six months before their birth. For GI.2, this maternally derived antibody may persist until 28 days of life [33].

Alternatively, the naturally acquired immunity could be from previous sub-lethal exposure to these viruses. Both suppliers had no recent reports of disease; but given that GI.1 does not generally cause disease in rabbits less than nine weeks of age, natural infection with these viruses cannot be excluded. GI.1a is known to circulate naturally in the geographic region from which these rabbits were obtained [19]. A cohort of these rabbits with GI.1 immunity were kept until 33 weeks of age. At this older age, antibodies were still detectable, confirming that immunity was not due to maternal antibodies. In addition, these rabbits had complete cross-protection against a substantial PO challenge (1500 RID_50_) with GI.1a at 33 weeks of age, suggesting that this immunity was not waning. Therefore, it is more likely that these rabbits acquired immunity due to natural exposure to GI.1 at farm 1.

Likewise, this same scenario could exist for the rabbits with GI.2 immunity from farm 2. GI.2 can cause disease in younger rabbits from four weeks of age, and the pathogenicity of Australian GI.2 variants is close to 100% in susceptible animals [20]. However, young rabbits with GI.2 maternally derived immunity are resistant to disease with GI.2 [21,35]. Experimental work using passively immunised kittens, to mimic the presence of maternal antibodies, demonstrated a robust antibody response after virus challenge [21].

Despite an extremely high 150,000 RID_50_ dose, all rabbits that had survived a GI.4cP-GI.2 infection did not succumb to a heterologous GI.1bP-GI.2 challenge. As both viruses share the same GI.2 capsid protein, it is expected that previous infection with GI.4cP-GI.2 should protect against a lethal challenge of GI.1bP-GI.2 [36]. This immune protection has implications for the use of the GI.2 virus (GI.1bP-GI.2) as a future biocontrol agent. GI.2 is currently the dominant genotype across Australia [19]. It is expected that GI.1bp-GI.2 would not be an effective biocontrol agent in a population of feral rabbits with pre-existing GI.2 immunity. Therefore, the immunity of this population should be established prior to intentional virus release when using RHDV as a biocontrol agent in feral rabbit populations, to achieve optimal population control.

The prototype inactivated GI.2-specific vaccine did not offer any cross-protection against GI.1a or GI.1c (Table 1 and Table 2) in contrast to those rabbits with either experimentally or naturally acquired immunity resulting from infection. This is consistent with previous studies where GI.1 vaccinal protection does not offer an acceptable level of cross-protection to GI.2 [11,15]. The lack of cross-protection from inactivated vaccines is likely due to a difference in immune response as observed with other viral infections [37]. Inactivated vaccines primarily stimulate humoral immunity, but virus infection generates a broader response including strong cell mediated and mucosal immunity [17]. In Australia, to protect domestic rabbits, a multivalent RHDV vaccine that offers complete protection against infection with GI.1a, GI.1c and GI.2 should be considered.

## 5. Conclusions

The results presented here demonstrate that, in laboratory rabbits, there is cross-protective immunity, even between heterologous strains, that protect against lethal disease. The extent of cross-protection conferred depends on the source of immunity (experimentally acquired, vaccinal protection, or naturally acquired), the virus genotype that has induced the immunity (GI.1a, GI.1c, GI.2 or GI.4c), the timing between acquisition of immunity and challenge, and in some cases the infectious dose of the challenge virus. Consequently, for biocontrol programs, the immunity of feral rabbit population should be established prior to intentional RHDV release. For optimal protection of domestic rabbits from unintentional infection with lagoviruses, a multivalent vaccine providing broad immunity to the recognized viruses should be considered.

## Figures and Tables

**Figure 1 vaccines-10-00666-f001:**
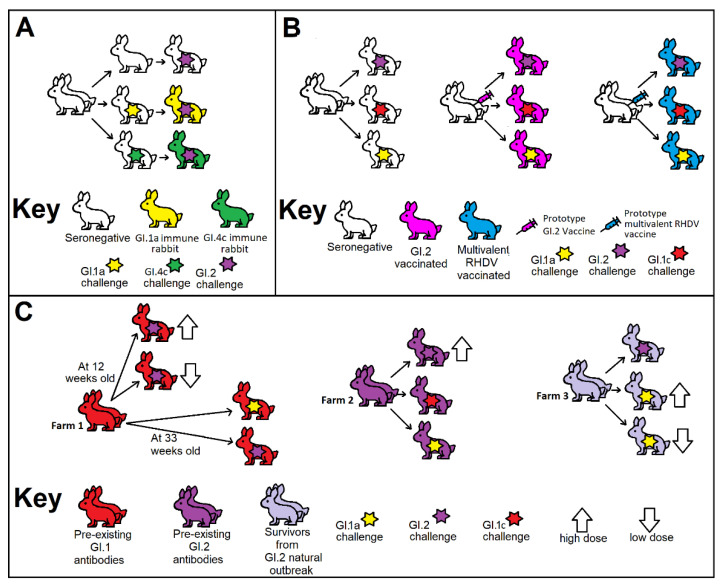
(**A**) Schematic diagram for the production and challenge of rabbits immune to GI.1a or GI.4c**.** Rabbits with antibodies against GI.1a or GI.4c were produced by initially infecting by the oral route of infection at five weeks of age. Rabbits were subsequently challenged orally with GI.1bP-GI.2 at 12 weeks of age. (**B**). Schematic diagram for the production and challenge of rabbits vaccinated with a prototype GI.2-specific vaccine or a multivalent RHDV vaccine. Rabbits were vaccinated via subcutaneous injection at 10–12 weeks of age, and subsequently challenged *per os* with heterologous RHDV variants 28 days after vaccination. (**C**) Schematic diagram for the challenge of rabbits with naturally acquired immunity from three commercial farms. Rabbits were challenged with various heterologous RHDV variants (at various ages and with various infectious doses).

**Figure 2 vaccines-10-00666-f002:**
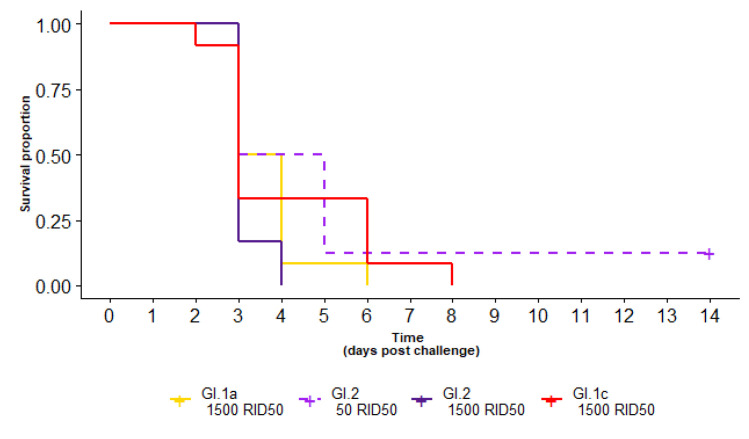
Survival curve of seronegative rabbits after challenge with GI.1c (red), GI.1a (yellow) or two different doses of GI.2 (purple-dashed, dark purple). For seronegative rabbits, there was no statistical difference in survival between the variant used nor the infectious dose.

**Figure 3 vaccines-10-00666-f003:**
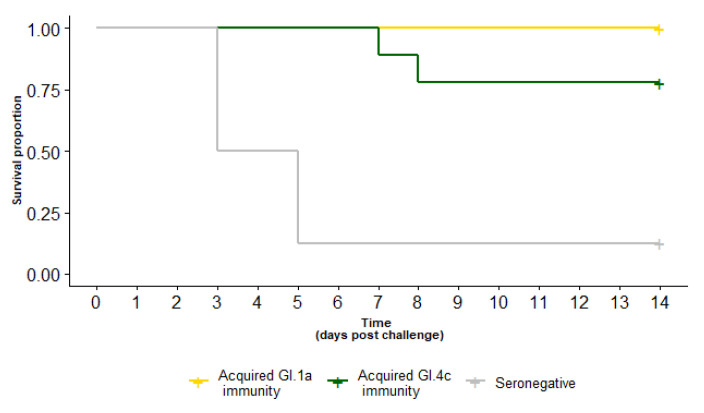
Survival curve of rabbits with experimentally acquired GI.1a (yellow) or GI.4c (green) immunity compared to seronegative (grey) rabbits after a GI.2 challenge (50 RID_50_ PO). Immunity with either GI.1a or GI.4c variants provided significant cross-protection against GI.2 challenge.

**Figure 4 vaccines-10-00666-f004:**
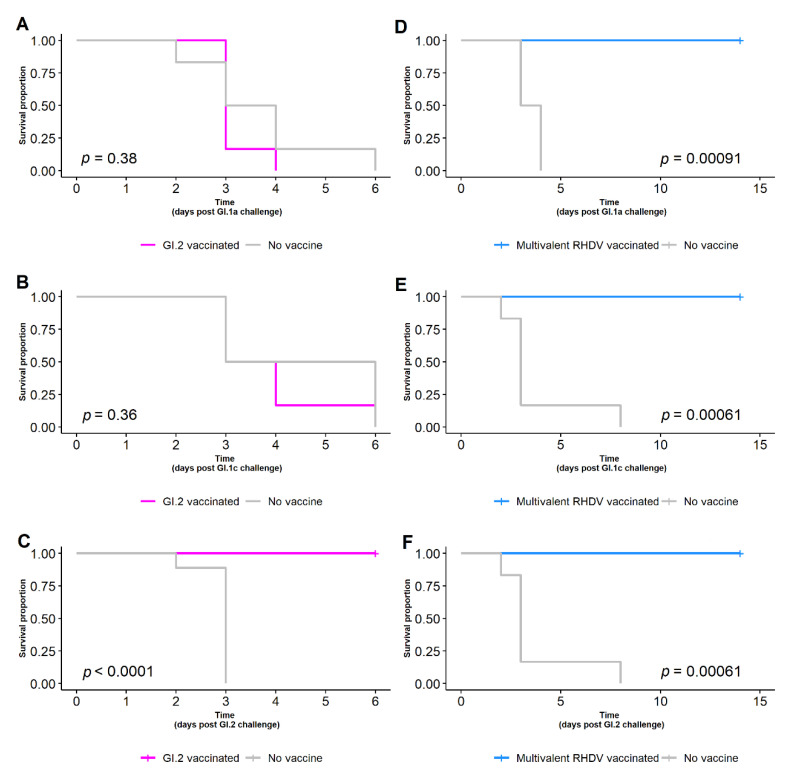
Survival curve of rabbits given a GI.2-specific vaccine (**A**–**C**, magenta), multivalent RHDV vaccine (**D**–**F**, blue), or no vaccine (**A**–**F**, grey) and challenged with 1500 RID_50_ of GI.1a, GI.1c or GI.2. While the GI.2-specific vaccine provided complete protection against GI.2 challenge, only the rabbits given the multivalent RHDV vaccine were protected against GI.1a, GI.1c and GI.2 challenge.

**Figure 5 vaccines-10-00666-f005:**
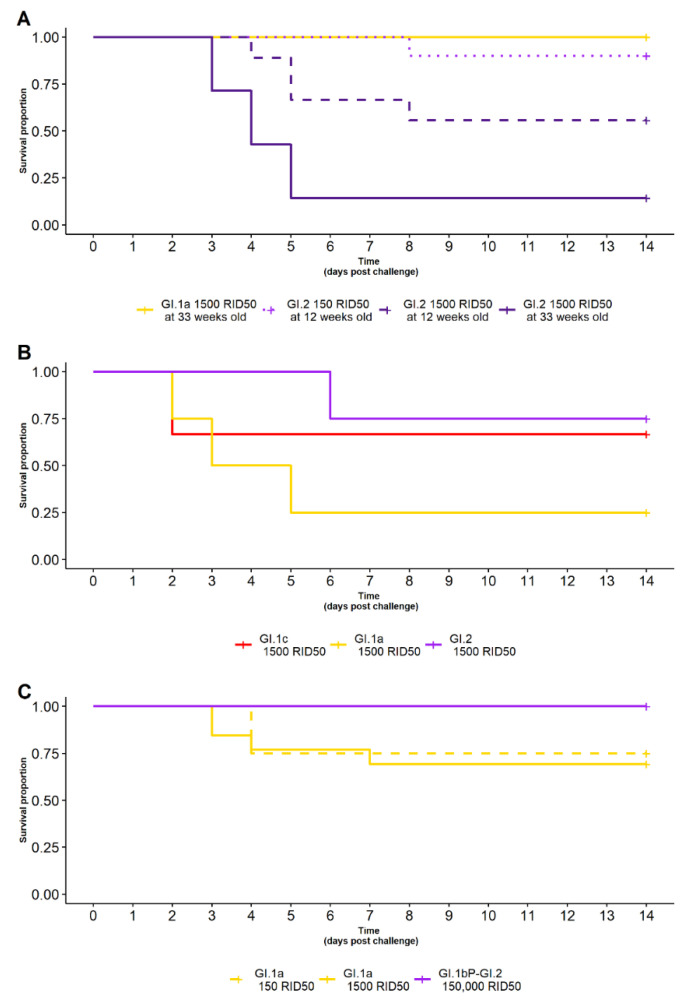
(**A**) Survival curve of rabbits with naturally acquired GI.1 immunity challenged with GI.1a (yellow), or with various doses of GI.2 (150 RID_50_—dotted-purple, 1500 RID_50_—purple) at different ages (12 weeks—purple dashed line, 33 weeks—purple solid line). All rabbits survived GI.1a challenge. A dose-dependent and age-related difference in survival was observed against GI.2 challenge, with rabbits given a higher dose or challenged at an older age less likely to survive. (**B**) Survival curve of rabbits with naturally acquired GI.2 immunity after GI.1c (red), GI.1a (yellow) or GI.2 (purple) challenge. Fewer rabbits survived a GI.1a challenge compared to the other challenge viruses. (**C**) Survival curve of rabbits with RHDV (GI.4cP-GI.2) immunity after heterologous GI.2 (purple) or GI.1a challenge at various doses (150 RID_50_—yellow-dashed, 1500 RID_50_—yellow-solid). All rabbits survived GI.2 challenge, whereas fewer rabbits survived GI.1a challenge.

**Table 1 vaccines-10-00666-t001:** Immunity status, age at challenge, challenge virus, infectious dose and the subsequent survival proportion of rabbits with experimentally or naturally acquired immunity.

Immunity Status	Age at Challenge (Weeks)	Challenge Virus	Infectious Dose (RID_50_ ^1^)	Survived/Total
Seronegative	11	GI.2 ^2^	50	1/8 ^3^
Seronegative	16	GI.1a	1500	0/12
Seronegative	16	GI.1c	1500	0/12
Seronegative	16	GI.2^2^	1500	0/12
GI.4c	11	GI.2^2^	50	7/9
GI.1a	11	GI.2^2^	50	7/7
GI.1 ^4^	12	GI.2^2^	150	9/10
GI.1 ^4^	12	GI.2^2^	1500	5/9
GI.1 ^4^	33	GI.2^2^	1500	1/7
GI.1 ^4^	33	GI.1a	1500	7/7
GI.2	10–12	GI.1a	1500	1/4
GI.2	10–12	GI.1c	1500	2/3
GI.2	10–12	GI.2^2^	1500	3/4
GI.2 ^5^	12	GI.1a	150	9/12
GI.2 ^5^	12	GI.1a	1500	9/13
GI.2 ^5^	12	GI.2 ^2^	150,000	12/12

^1^ The 50% rabbit infectious dose (RID_50_). ^2^ This GI.2 refers to a naturally occurring RHDV2 recombinant virus designated GI.1bP-GI.2 (GenBank acc. Number MW467791). ^3^ This one surviving rabbit did not seroconvert after challenge and was not infected with GI.2. ^4^ These rabbits could have antibodies from either GI.1a or GI.1c exposure. ^5^ This GI.2 refers to a naturally occurring RHDV2 recombinant virus designated GI.4cP-GI.2 (GenBank acc. Number MW460156).

**Table 2 vaccines-10-00666-t002:** Proportion of vaccinated rabbits that succumbed to rabbit hemorrhagic disease following oral inoculation with Australian lagoviruses.

Vaccination Status	Vaccine Dose (HAU ^1^)	Challenge Virus (Dose 1500 RID_50_)	Survived/Total
Seronegative	No vaccine	GI.1c	0/12
Seronegative	No vaccine	GI.1a	0/12
Seronegative	No vaccine	GI.2 ^2^	0/12
GI.2 ^2^	128	GI.1c	0/6
GI.2 ^2^	128	GI.1a	0/6
GI.2 ^2^	100	GI.2 ^2^	9/9
Multivalent	96	GI.1c	6/6
Multivalent	96	GI.1a	6/6
Multivalent	96	GI.2 ^2^	6/6

^1^ Total hemagglutination units. ^2^ This GI.2 refers to a RHDV2 recombinant designated GI.1bP-GI.2.

## Data Availability

All supporting data are included in this manuscript.

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
