# Peer review of "Immunological Cross-Protection between Different Rabbit Hemorrhagic Disease Viruses—Implications for Rabbit Biocontrol and Vaccine Development"

_vaccines, 2022, doi:10.3390/vaccines10050666_

Round 1

Reviewer 1 Report

Read suggestions in the attached PDF file.

Author Response

Dear Editorial Office,

Thank you for giving us the opportunity to submit a revised draft of our manuscript (vaccines-1680218) entitled “Immunological cross-protection between different strains of rabbit haemorrhagic disease virus – implications for rabbit biocontrol and vaccine development” by Tiffany W. O’Connor, Andrew J. Read, Robyn N. Hall, Tanja Strive, Peter D. Kirkland. We appreciate the time and effort that you and the reviewers have dedicated to providing valuable feedback to improve our manuscript.

We have been able to incorporate changes to reflect most of the specific suggestions provided by the reviewers. We have highlighted the changes within the manuscript in yellow. Detailed below is a point-by-point response to the reviewers’ comments.

We are grateful to both reviewers for these insightful comments on our manuscript. We look forward to hearing from you in due time regarding our submission.

Yours sincerely

Tiffany O’Connor

Comments from Reviewer 1

Comment 1. Page 1, line 17 - Delete one parenthesis in ((1)

Response: We agree that aesthetically, the double parenthesis is not appealing. To maintain the correct syntax we have also deleted the parentheses for following listed numbers too: 2., 3. and 4.

Comment 2. Page 1, line 21 - “GI.2 recombinant” is this a recombinant virus? Or a genetically modified genome virus?? make clear

Response:  Thank you for this suggestion, we have removed recombinant from the abstract and have clarified what we mean by recombinant in the introduction (line 42 in the revised manuscript). GI.1bP-GI.2 is a naturally occurring recombinant GI.2 virus.  

Comment 3. Page 1, line 24: Make clear “these interactions”, which interactions?? The immune status and the virus variants used for challenge? Or variant and doses, age or any other condition??

Response:  Thank you, we agree that this sentence is vague. We have replaced this final sentence to better describe the implications of the cross-protection observed in this study for biocontrol programs and vaccination of domestic rabbits in Australia (lines 23-27 in the revised manuscript).

Comment 4. Page 3, lines 111-117: This paragraph is confusing. Explain why to use RID50 units to inoculate a five week old rabbit group, and “2300 genome copies” to inoculate another group. If there is an equivalence, use RID50 or “genome copies” for both. Make clear somewhere that you are using virus preparations, containing VP10 (MF598301) and VP60 (KX357705.1) genes. It is very important to use infectious doses (RID) instead “genome copies”, since RID is the doses to infect rabbits. On the other hand, “genome copies” are not  infectious. Explain if there is an equivalence between, “RID50” and “genome copies” to avoid confusion.

Response: We agree with this comment. There is no real equivalence measure between RID50 and genome copies since this varies considerably between the different viruses. For clarification, the text has been updated (line 122, lines 126-128, and lines 131-134 in the revised manuscript).

Comment 5. Page 4, line 133: “Rabbits were not tested for anti-GI-4” and “these rabbits could possibly have had anti-GI. Antibodies” do not sound very scientific when you analyze the inmune cross reactions of variant viruses. Explain.

Response:  We thank the reviewer for this comment and would like to highlight that the series of experiments on these rabbits with naturally-acquired immunity were serendipitous. This has been clarified in the revised text (lines 146-148). While the nature of this ‘naturally-acquired’ immunity may not be precise; controlled, experimental infection in these rabbits offers an opportunity to demonstrate cross-protection in a population more representative of the ‘general’ domestic rabbit in Australia (in contrast to the laboratory rabbits with experimentally-acquired or vaccinal immunity).

Comment 6. Page 6, line 218: Define HI for the first time.

Response:  Thank you for noticing this abbreviation. This has been amended in the text (line 228).

Comment 7. Page 7, lines 251-252: Use this info (and similar info) to support your conclusions, this paper is focused on “Immunological cross protection….” And this is all about it. Include what is the best vaccine candidate you recommend to mitigate the problem, based on its immunological performance

Response:  We thank the reviewer for this comment. We feel that this would better belong in the discussion (lines 381-390), rather than the results section. Recommendation of a multivalent vaccine is also emphasised as the final sentence in our conclusion.

Comment 8. Page 13, lines 482 and 485: Be consistent with citation of references. V.gr. in line 482: “Gal, A.; Hofman, B.; …etc.” while in line 485 the authors use a different style: “Quin S, Underwood D, …etc.” Check out the punctuation all over the references and use it consistently.

Response:  Thank you for your attention to this citation. This has been corrected as suggested.

Comment 9: The paper contain relevant information, however, the selection of variables and technology used in the experiments, are not always consistent with the objectives of the experiments. For instance, different types of ELISA are used in order to detect antibodies against different RHDV strains, I would use the same methodology for all my experiments, so that, I can compare the results.

Response:  We thank the reviewer for this comment. Specific ELISAs are required to detect antibodies against different RHDV genotypes. Therefore, to determine serological status in our experiments, different specific ELISAs were used to detect antibodies against the different RHDV strains.

As suggested by reviewer 2, we have moved Table 1 and amended the caption of this table. This should better help a reader understand the explanatory variables investigated (immunity status, age at challenge, challenge virus, infectious dose) in rabbits with experimentally or naturally-acquired immunity and so the results may be more easily compared.

Comment 10: Some graphs showing the Optical Density obtained from the ELISA antibody detection results over the time, for each experimental group immune status, would give to the reader a better description about the amount of antibodies, protection and cross reaction (indirectly), before and after the challenge. However, the authors do not show those types of results or graphics.

Response:  Thank you for this comment. Full analysis of the magnitude of antibody responses and how this correlates with protection was beyond the scope of this study. We have decided to focus on presenting the survival proportion because these results are a better indicator as a direct measure of disease resistance.

Comment 11: There are two important results that can be obtained out of these experiments, the mortality rate after the challenge and antibody titer by ELISA, based on these two variables, the authors can make predictions on the protective cross reactions, I suggest to use this info in order to support the conclusions.

Response:  We thank the reviewer for this comment. Antibody titres have been poorly correlated to protection and or survival against RHDV. Therefore, as discussed in response to comment 10, a survival analysis is more relevant as a direct measure of cross-protection.  

Comment 12: Conclusions are based on general immunology concepts rather than experimental data: V. gr. we do not need to carry out an experiment to state that “the degree of cross protection depends on the virus that has induced the immunity….” Specially if we are focused on variants from the same virus. I highly recommend to use results to support the conclusions and do not forget to include what is the best candidate you recommend to mitigate the problem

Response:  We have amended the text to highlight that our conclusion is a succinct summary, listing the variables investigated, of the results from the series of experiments (lines 392-394 in the revised text).

Comment 13: For further research, consider to use a reverse vaccinology or vaccinomic approach to design or identify protective antigens and obtain more consistent results.

Response:  We thank the reviewer for this comment and will take this on as a future consideration.

Reviewer 2 Report

This paper talks about cross protection and immunity in rabbits in Australia, which is a topic with high significant in ecological equilibrium. The overall flow of the paper is smooth and logical. The only thing I would suggest is to make a table for your experimental design since you have different viruses like GI.1a, GI.1c, GI.2, GI.4 ... which could be confusing. A table would help a lot to make things clear.

Author Response

Comments from Reviewer 2

This paper talks about cross protection and immunity in rabbits in Australia, which is a topic with high significant in ecological equilibrium. The overall flow of the paper is smooth and logical. The only thing I would suggest is to make a table for your experimental design since you have different viruses like GI.1a, GI.1c, GI.2, GI.4 ... which could be confusing. A table would help a lot to make things clear.

Response: Thank you for your encouraging comments. As suggested, to help clarify the experimental design, we have moved Table 1 so that it appears earlier in the manuscript and amended the caption of this table.